| Open Peer Review | Human Microbiome | Methods and Protocols
# Evaluating stool microbiome integrity after domestic freezer storage using whole-metagenome sequencing, genome assembly, and antimicrobial resistance gene analysis

Paula Momo Cabrera,[1,2] Nicholas A. Bokulich,[2] Petra Zimmermann[1,3,4,5]

**ABSTRACT** The gut microbiome is crucial for host health. Early childhood is a critical period for the development of a healthy gut microbiome, but it is particularly sensitive to external influences. Recent research has focused on using advanced techniques like shotgun metagenome sequencing to identify key microbial signatures and disruptions linked to disease. For accurate microbiome analysis, samples need to be collected and stored under specific conditions to preserve microbial integrity and composition, with −80°C storage considered the gold standard for stabilization. This study investigates the effect of domestic freezer storage on the microbial composition of stool samples obtained from 20 children under 4 years of age with the use of shotgun metagenome sequencing. Fresh stool samples were aliquoted into sterile tubes, with one aliquot stored at 4°C and analyzed within 24 hours, while others were frozen in domestic freezers (below −18°C) and analyzed after 1 week, 2 months, and 6 months. Assessments of contig assembly quality, microbial diversity, and antimicrobial resistance genes revealed no significant degradation or variation in microbial composition.

**IMPORTANCE** Most prior studies on sample storage have relied on amplicon sequencing, which is less applicable to metagenome sequencing—given considerations of contig quality and functional gene detection—and less reliable in representing microbial composition. Moreover, the effects of domestic freezer storage for at-home stool collection on microbiome profiles, contig quality, and antimicrobial resistance gene profiles have not been previously investigated. Our findings suggest that stool samples stored in domestic freezers for up to 6 months maintain the integrity of metagenomic data. These findings indicate that domestic freezer storage does not compromise the integrity or reproducibility of metagenomic data, offering a reliable and accessible alternative for temporary sample storage. This approach enhances the feasibility of large-scale at-home stool collection and citizen science projects, even those focused on the more easily perturbed early life microbiome. This advancement enables more inclusive research into the gut microbiome, enhancing our understanding of its role in human health.

**KEYWORDS** microbiota, fecal, gut, infant, shotgun sequencing, −20°C

The composition of the stool microbiome has been linked to many diseases, including allergic, inflammatory, rheumatological, metabolic, and psychiatric diseases (1–5). To accurately determine dysbiotic or suboptimal microbiome states and identify confounding factors, longitudinal sampling of large, representative population cohorts is essential. However, this can be logistically challenging. The current gold standard approach for preserving microbiome integrity is immediate DNA extraction or freezing of the stool sample at −80°C as the addition of stabilization buffers can affect DNA quantity and

**Peer Reviewers** Elliot S. Friedman, University of Pennsylvania, Philadelphia, Pennsylvania, USA; Maria Ingrid Cecilia Rubin, Hvidovre Hospital, Hvidovre, Denmark

Address correspondence to Petra Zimmermann, petra.zimmermann@unifr.ch.

Nicholas A. Bokulich and Petra Zimmermann contributed equally to this article.

The authors declare no conflict of interest.

See the funding table on p. 11.

purity or lead to bacterial cell lysis (6, 7). While a number of studies have now tested refrigeration and room-temperature storage conditions (8–14), the impact of freezer temperature on microbiota composition during storage warrants further attention.

Immediate DNA extraction is not always feasible, particularly in studies including geographically dispersed participants. Studies using 16S rRNA amplicon sequencing methods show that DNA integrity and microbial composition are maintained with storage of the stool sample at room temperature for up to 24 hours (8, 15) , at 4°C for up to 24–72 hours (15–18), and at −20°C for up to 7 days (15, 18, 19). Conversely, other studies indicate that storage of the stool sample at room temperature over 12–72 hours (9, 16, 20) and at −20°C for 3–7 days (21) results in significant changes in the bacterial composition. One study has even reported changes in the relative abundance of bacterial phyla after just 30 minutes of exposure to room temperature (21). Recent studies suggest that storage of the stool sample at −20°C may suffice for short-term preservation (22), challenging the necessity of −80°C freezing. Thus, it remains uncertain whether the widely accepted gold standard of −80°C freezing is truly necessary, offering a possibility for investigating alternative cost-effective and accessible storage solutions.

A key concern with using domestic freezers is the occurrence of freeze–thaw cycles, particularly in frost-free freezers. These cycles cause periodic temperature fluctuations, which have been linked to changes in microbiome composition in some studies (19, 21), but not in others (18). These differences may reflect variability in sample sizes ($n$ = 11, 4, and 3) and study design. Most modern household refrigerator-freezers are equipped with automatic frost removal systems that operate in cycles lasting between 10 and 30 minutes (23, 24), typically occurring once or twice daily. During these defrost cycles, the air temperature in the freezer can increase to approximately −4°C (25), which could potentially impact the stability and preservation of the microbial community.

In this study, we investigated the stability of microbial composition in stool samples obtained from young children, which are stored in home freezers (−18°C) over 6 months, a condition that has not been previously investigated and could pose significant logistical advantages to current sample collection and storage methods.

Enabling participants to store samples in their home freezers can reduce the frequency of nurse visits for sample collection, thereby lowering personnel and travel costs, as well as the logistical burden on participants and study coordinators. This becomes particularly relevant as the field moves toward larger cohort sizes and higher temporal density in microbiome samples.

## RESULTS

### Stool microbial diversity

To investigate the stability of microbial diversity in stool samples from our cohort of 20 children stored in home freezers over a 6-month period, we analyzed read-based metagenomic data from samples collected at four distinct time points: week 0 (0W), week 1 (1W), month 2 (2M), and month 6 (6M). Principal component analysis on Aitchison distances (Fig. 1a) indicated that storage at −20°C had a minimal impact on the overall microbial community structure as no clustering by storage duration was observed. The mean Aitchison distance, which represents compositional differences between samples at different time points, showed no significant variations (Fig. 1b). Bray–Curtis and Jaccard distance metrics (Fig. S1a and b respectively), further showed that inter-individual variability surpasses temporal variability. As illustrated in Fig. 1c, alpha diversity remained stable over time, with no significant differences observed between time points, not even in comparison to the unfrozen samples at 0W. Additionally, the number of observed species (Fig. 1c) showed no significant differences across time points.

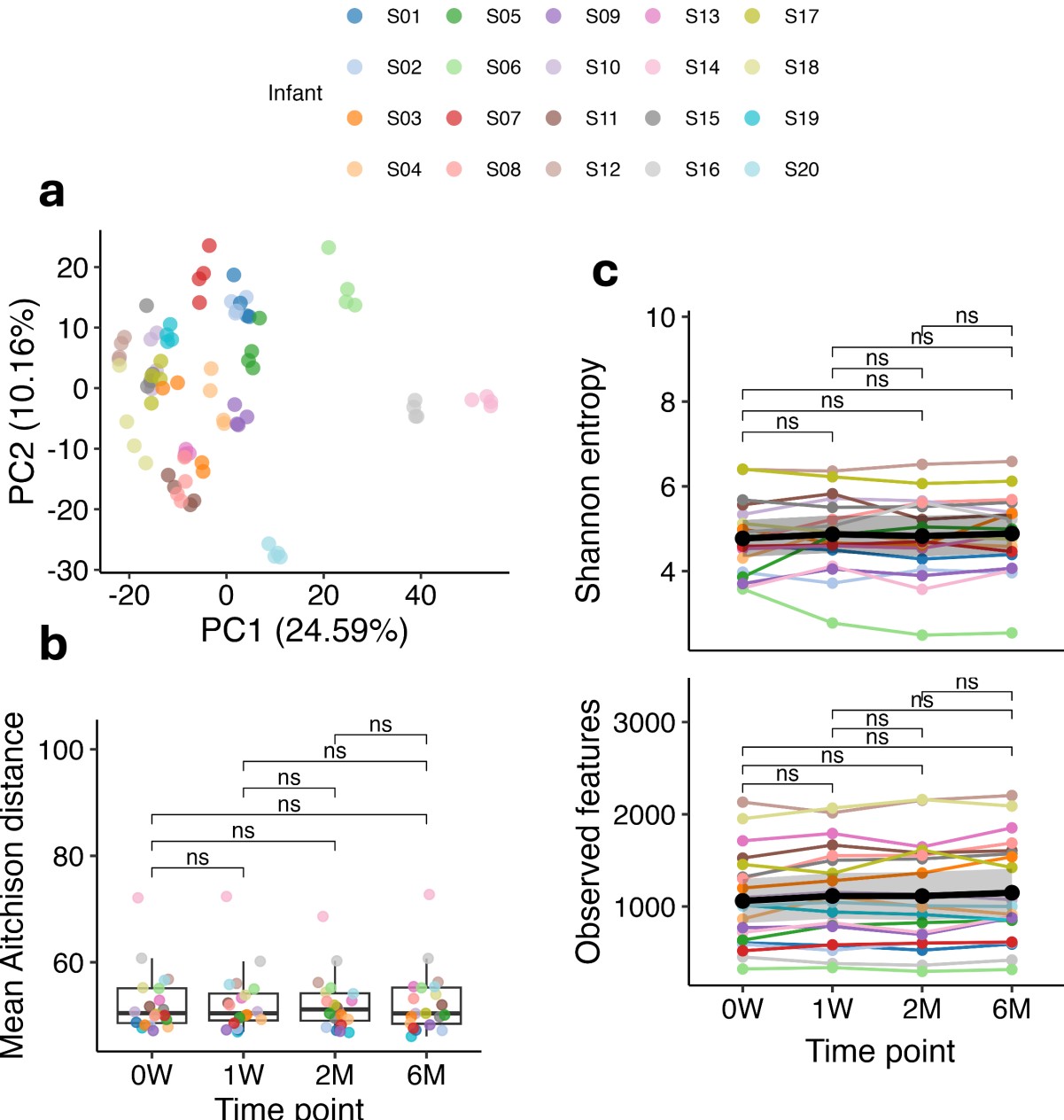

**FIG 1** (a) PCoA based on the Aitchison distance matrix, illustrating the beta diversity of the microbial communities. Each dot represents a sample, colored by child ID (S01–S20). (b) Mean Aitchison distance between samples at different time points, representing compositional differences. The mean distance is calculated as the average pairwise Aitchison distance for all samples within each time point, with individual dots representing the pairwise distances between samples. (c) Alpha diversity metrics across time points: the Shannon diversity index (top panel) and observed features (bottom panel) are shown at 0 weeks (0W), 1 week (1W), 2 months (2M), and 6 months (6M). The Shannon index reflects diversity within samples, while observed features indicate species richness. Thick black lines represent the mean values, with shaded gray areas denoting standard deviation. The Wilcoxon rank-sum test with Bonferroni correction was used to assess differences, with no significant differences (ns) observed between the time points for panels b and c.

## Inter-individual differences have a greater influence on stool microbial diversity than temporal effects

To better understand the factors influencing the microbial community structure, we employed linear mixed effects (LME) models (Table S1). Our analysis revealed that storage time did not significantly affect microbial community composition when evaluated using Aitchison ($P = 0.267$, $\beta = -0.03$) and Jaccard ($P = 0.836$, $\beta = 0.000$)

metrics, although a weak but significant effect was observed with Bray–Curtis ($P = 0.007$, $\beta = -0.004$).

In contrast, children's age emerged as a significant factor, consistently reflecting inter-individual variation in microbial communities. This was particularly evident in the Aitchison ($P = 0.005$, $\beta = 3.062$) and Jaccard ($P = 0.004$, $\beta = 0.019$) metrics, although the Bray–Curtis metric did not show a significant effect ($P = 0.629$, $\beta = 0.006$).

Further supporting the minimal impact of temporal changes, PERMANOVA results indicated no significant differences in beta diversity across different time points ($P = 1$ for Aitchison, $P = 0.935$ for Bray–Curtis, and $P = 0.992$ for Jaccard) (Table S2).

To evaluate the relative contributions of temporal versus inter-individual variation in stool microbiome profiles, we implemented a random forest classifier to predict the sampling time points. The classifier's performance was suboptimal, as evidenced by the confusion matrix (Fig. S2a), where no clear pattern of accurate classification emerged. The ROC curves (Fig. S2b) further illustrate the classifier's inefficacy, with area under the curve (AUC) values across different time points ranging from 0.10 to 0.24, only marginally above the expected performance of random guessing. Notably, the overall accuracy failed to exceed random chance, emphasizing the difficulty in distinguishing samples based solely on their microbiome profiles over time (Fig. S2c).

## Temporal dynamics of specific taxa

A total of 115 species were detected at or above 1% relative abundance among all samples; however, none showed significant deviations from the pre-freezing baseline (0W) following ANCOM-BC analysis. As can be observed in Fig. 2, subtle qualitative fluctuations were noted among subjects over time, and these changes in the microbial community did not reach statistical significance neither at the genus or species level.

## Temporal variability in antimicrobial resistance

The longitudinal stability of AMR genes detected at 0W was assessed using AMRFinder-Plus and RGI annotations (Fig. S3). Both tools identified clinically relevant AMR genes. Most genes detected at 0W remained consistently present across 1W, 2M, and 6M, demonstrating robust preservation under varying storage conditions.

Genes such as *tet(Q)* were consistently detected across all time points by both tools, highlighting their agreement in tracking highly prevalent AMR genes. However, some differences were observed between the tools: AMRFinderPlus focused on clinically significant genes and resistance mechanisms (e.g., *erm(B)* and *cfxA*), whereas RGI

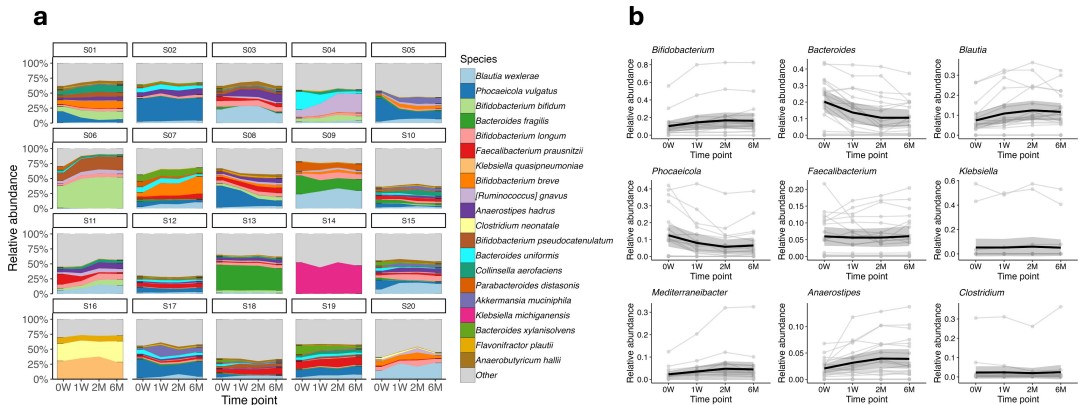

**FIG 2** (a) Stacked area plots showing the relative abundance of the top 20 most abundant species across different time points (0W = week 0, 1W = week 1, 2M = month 2, and 6M = month 6) for each child (S01–S20). Each color represents a different species, illustrating the dynamic changes and stability in the microbial community composition over the first 6 months of life. (b) Line plots depicting the relative abundance trends of the most abundant genera. Each plot shows the relative abundance over time per individual child data (thin lines) and the overall mean trend (thick line).

annotated a broader range of resistance genes, including those associated with efflux pumps (e.g., *mdtA* and *acrB*).

To further explore AMR gene dynamics, changes in gene abundance were examined using read-based alignment to the CARD database via the RGI application. In Fig. 3a, the PCoA based on the Aitchison distance shows variability in AMR profiles across different subjects and time points. The first principal component (PC1) accounts for 25.82% of the variance, while the second principal component (PC2) explains 11.13%. To further understand the drivers of these observed changes, we analyzed the metadata variables contributing to the principal coordinates from the PCoA of Jaccard distance (Fig. S4). Specifically, time point (week) ($R^2$=0.04 for PC1 and $R^2$=0.013 for PC2) was demonstrated to be poorly correlated, consistently revealing the lack of impact of sample collection time points on AMR profiles. Conversely, child age (months) was the most substantial contributor to the observed variability ($R^2$=0.54 for PC1 and $R^2$=0.32 for PC2).

Furthermore, Fig. 3b displays antimicrobial resistance profiles over time. The analysis highlighted that the most prevalent AMR-conferring genes detected, including "B. adolescentis rpoB Rif^R" (mutated RNA polymerase β subunit (*rpoB*) conferring resistance to rifampicin) and "B. bifidum ileS Mup^R" (mutated isoleucyl-tRNA

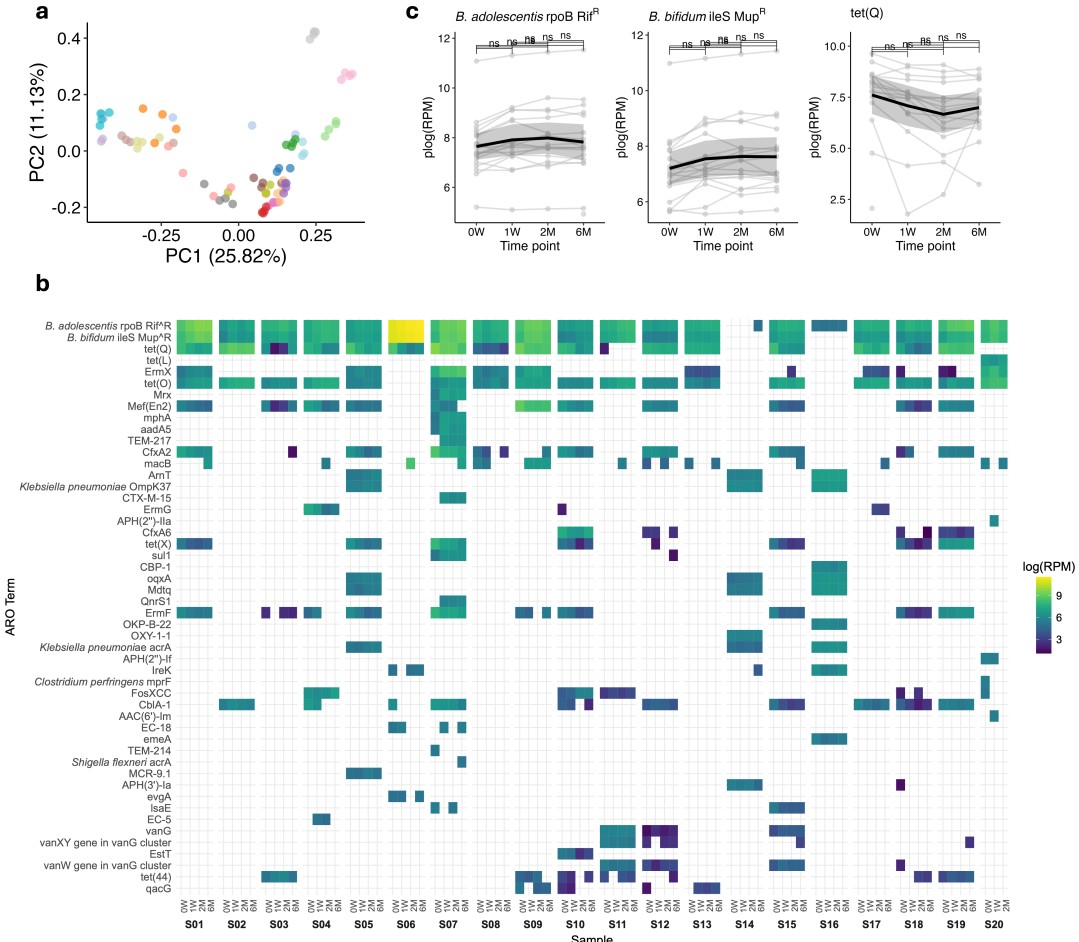

**FIG 3** (a) Principal coordinate analysis (PCoA) plot depicting beta diversity based on the Jaccard distance matrix, calculated from antimicrobial resistance (AMR)-conferring genes. Each point represents a sample, colored by child (S01–S20), as indicated in the legend, with the percentage of variance explained by PC1 and PC2. (b) Heatmap of the top 20 most abundant ARO terms, displaying log-transformed normalized reads per million (RPM) values. (c) Line plots depicting the relative abundance trends of the most abundant AMR-conferring genes. Each plot shows RPM over time per individual child data (thin lines) and the overall mean trend (thick line). The y-axis is displayed on a pseudo-log scale (plog, log(1 + x)) to enhance visualization of small or zero values. The Wilcoxon rank-sum test with Bonferroni correction was used to assess differences, with no significant differences (ns) observed between the time points (0W = week 0; 1W = week 1; 2M = month 2; 6M = month 6).

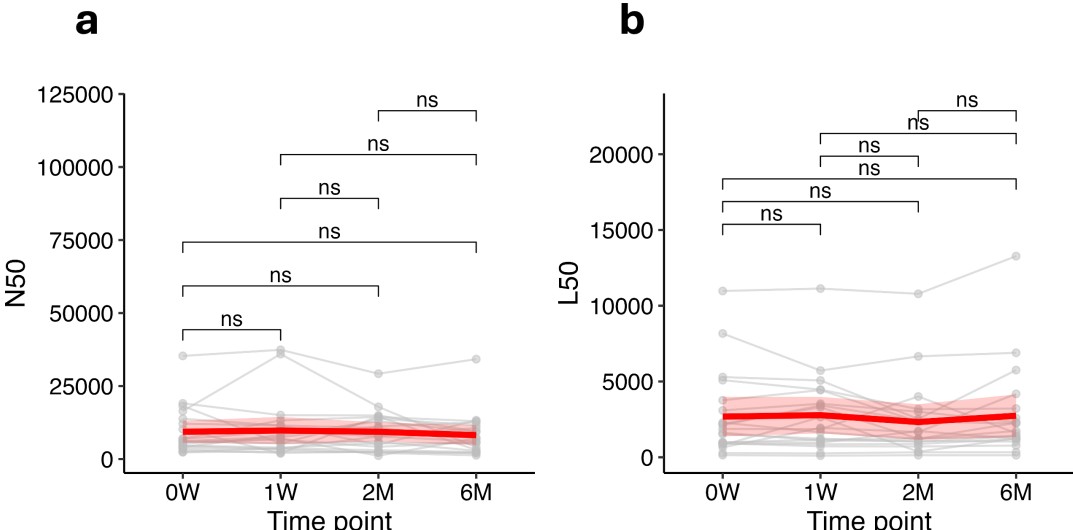

**FIG 4**  N50 (a) and L50 (b) metrics in stool microbiome samples stored over time (0W = week 0, 1W = week 1, 2M = month 2, and 6M = month 6). Line plots show individual child data (gray lines) and the overall mean trend (red line), with shaded red areas representing the standard deviation. The Wilcoxon rank-sum test with Bonferroni correction was used to compare metrics across time points, with no significant differences (ns) observed.

synthetase gene (*ileS*), conferring resistance to mupirocin), showed stable mean abundance overtime across all children. In contrast, "tet(Q)" (tetracycline-resistant ribosomal protection protein gene) showed a trend of decreasing abundance over time (Fig. 3c). However, none of this fluctuations resulted to be statistically significant (Fig. 3d).

## Contig assembly quality over time

Next, we evaluated the stability of contig assembly quality in stool microbiome samples over time to assess whether domestic freezer storage could induce DNA damage or other changes impacting assembly quality. We assessed key metrics such as N50 and L50 across four different time points: week 0 (0W), week 1 (1W), month 2 (2M), and month 6 (6M). These metrics are critical indicators of the quality and completeness of genome assemblies obtained from metagenomic sequencing data, representing the length of the shortest contig at the 50% genome assembly threshold (N50) and the number of contigs whose lengths sum to 50% of the genome assembly (L50).

The median N50 values show slight fluctuations across the different time points, with no significant differences observed (ns) between 0W, 1W, 2M, and 6M. (Fig. 4a). Similarly, the L50 values across the same time points also exhibited minimal variation, with no significant differences (ns) detected between the different storage durations. (Fig. 4b).

## DISCUSSION

In the rapidly evolving microbiome field, efforts have increasingly been directed toward larger cohort sizes and higher temporal density in sample collection. In this context, the need for reliable, cost-efficient, and low-burden sample collection and storage solutions have become paramount. Ensuring that storage conditions do not introduce significant biases or artifacts in metagenome sequence data is crucial for accurately tracking functional gene composition and microbial community dynamics over time.

Our findings demonstrate that DNA integrity, microbial and resistance-conferring gene diversity, as well as contig assembly quality remain stable over a 6-month storage period in home freezers. By employing shotgun metagenomic sequencing, we achieved species-level resolution in our analysis, allowing for a more comprehensive assessment of microbial diversity that extends beyond the conventional focus on taxonomic stability during storage. This provides a holistic view of stool DNA stability under storage in home freezers, offering valuable insights for optimizing long-term storage protocols.

Additionally, the study's design, which included multiple time points, adds robustness to our findings.

In this study, the core microbiome retains its richness and evenness over time, irrespective of the increasing number of freeze–thaw cycles endured during storage in home freezers. While contig assembly results were an essential quality check, our study used read-based analyses, such as microbial diversity and AMR profiling, which provide more detailed insights into the microbial community structure and function. While the relative abundances of species and AMR genes may fluctuate with storage in home freezers, the overall community structure remains largely stable. In fact, similarly to what was observed by Ilett et al. (26), our study found no significant differences between non-frozen 0 W samples and the frozen 1W up to 6 M samples. However, specific microbial taxa which were found to have subtle temporal fluctuations in abundance included *Bacteroides* spp., whose relative abundance decreased, and *Bifidobacterium* spp., the relative abundance of which increased over the 6-month storage period. These taxa-dependent variations are consistent with previous studies and could be attributed to differences in bacterial cell wall structure, metabolic activity, and resilience to freezing and thawing processes (22, 27–29). These findings highlighted the intricate dynamics within microbial communities and the importance of considering taxa-specific responses when interpreting microbiome data. Our study identified both high stability in the most abundant and frequent resistance-conferring genes and subtle shifts in those less abundant and highly infrequent. These shifts are likely more related to the heterogeneity of the samples than to storage conditions, emphasizing the complex and dynamic nature of microbial communities. The prevalence of highly infrequent and sparse AMR-conferring genes suggest that while some resistance-conferring genes maintain a consistent presence, others appear to be more sporadic and sparse. These findings aligned with those of previous research (27), which demonstrated that different storage methods could influence the detection and quantification of AMR-conferring genes, impacting the interpretation of resistance profiles in microbial communities and underscored the importance of longitudinal monitoring of AMR profiles to understand the evolution of resistance genes. Although our study observed sparse AMR genes in this cohort of healthy Swiss children under 4 years of age, prior research indicates that the gut often harbors a higher abundance of ARGs in early life compared to adulthood (30, 31). This is likely due to early colonization by antibiotic-resistant bacteria from maternal and environmental sources, with ARG levels declining as the microbiome matures. To comprehensively understand AMR dynamics, longitudinal studies across diverse cohorts, including adults and elderly populations, are essential to capture age-specific patterns influenced by antibiotic exposure and microbiome development. Our study addressed a significant gap in the literature by demonstrating the feasibility of using home freezer storage for preserving stool samples. While different studies have investigated the temporal variability of a microbial community under different storage conditions, validating the stability of stool DNA stored in home freezers offers significant advantages for large-scale cohort studies, particularly those involving long-term follow-up and geographically dispersed participants. Notably, access to ultra-low temperature storage facilities (−80°C) is limited in many regions. By reducing the need for specialized storage facilities, researchers can optimize resource allocation and reduce logistical burdens on participants and study coordinators. This finding is particularly relevant for large-scale and citizen science projects, where cost-effective and accessible storage solutions are essential to promote inclusivity and broader participation in research (7).

Nevertheless, our study has several limitations. First, our study focused on a 6-month storage period. Future research should explore longer storage durations to assess longer-term storage effects. Additionally, our study included a relatively small sample size of 20 children, which may limit the generalizability of the results. A larger sample size would provide more robust data and allow for a more comprehensive analysis of variability across different individuals. While we employed ANCOM-BC to identify differentially abundant taxa at both the species and genus levels and observed no

statistically significant differences over time, this approach relies on relative abundance data and does not capture absolute abundance. Future studies incorporating methods such as cell quantification (e.g., qPCR or flow cytometry) or spike-in standards during sequencing could provide deeper insights into temporal fluctuations in absolute abundance, which may reflect ecological dynamics not fully captured by the metrics used in this study. Future research should also include a broader range of sample types and participant cohorts such as individuals with gastrointestinal diseases (e.g., *Clostridioides difficile* infections) or those colonized by multiresistant organisms. These populations often harbor higher abundances and greater diversity of AMR genes due to increased antibiotic exposure and altered microbiomes, making them particularly relevant for assessing microbial composition and AMR gene stability under varying storage conditions. This will strengthen the applicability of findings for large-scale microbiome studies and clinical applications.

## Conclusion

The finding that home freezers can be used to effectively store stool samples for microbiome analysis significantly enhances the feasibility of long-term studies that involve at-home collection of stool samples. This approach promotes broader participation by allowing participants to conveniently store samples at home. Our results support the use of home freezer storage as a viable alternative to conventional methods, ensuring reliable and reproducible results. This advancement facilitates robust research into microbial dynamics and disease mechanisms. Future research should further explore the long-term stability of samples under various conditions to optimize preservation protocols and advance microbiome research.

## MATERIALS AND METHODS

### Stool sample collection and storage in home freezers

Fresh stool samples from 20 healthy Swiss children (55% female) with a mean age of 22.4 (range 2 to 56) months were aliquoted into four Fecon sterile tubes (Fecotainers, Medical Wire & Equipment Co. Ltd, United Kingdom). One aliquot was stored at 4°C and extracted and analyzed within 24 hours (0W), while the other three aliquots of the same sample were frozen in domestic freezers (below −18°C) and analyzed after 1 week (1W), 2 months (2M), and 6 months (6M). The domestic freezers used in this study included several models, such as V-ZUG Classic eco, Bosch Serie 6 NoFrost, Indesit Class TZAAA10.1, Liebherr MedLine, and Electrolux SG 164. The Bosch Serie 6 NoFrost and Liebherr MedLine freezers were frost-free models with automatic defrost cycles, while the V-ZUG, Indesit, and Electrolux models required manual defrosting. Samples were kept under the same conditions during transport to the Microbiota and Children Laboratory at the University of Fribourg, Switzerland.

### DNA extraction and metagenomic shotgun sequencing

Aliquots of stool samples were thawed at room temperature, and 100 mg of each sample was used for DNA extraction using the FastDNA SPIN Kit for Soil (MP Biomedicals, Illkirch-Graffenstaden, France) according to the manufacturer's instructions. DNA concentrations were measured using Qubit dsDNA High Sensitivity Assay kits (Life Technologies, California, United States). The DNA in the negative control was below the detection limit (<0.01 ng/µL) and was not sequenced. Sequencing libraries with an insert size of approximately 600 bp were prepared using Nextera DNA Flex library preparation kits (Illumina, San Francisco, United States), with the addition of Illumina PhiX DNA. Paired-end sequencing (2 × 149 bp) was performed on a NextSeq 550 system (Illumina) using high-output flow cells. Positive controls included bacterial and fungal mock communities (Gut Microbiome Whole Cell Mix MSA-2006 and Mycobiome Whole Cell Mix, respectively, ATCC, Manassas, United States).

## Bioinformatic analyses

### Quality filtering and removal of human-mapping reads

FASTQC (v.0.11.9) was used to assess the quality of raw reads (32). Low-quality reads were filtered and trimmed using Trimmomatic v0.39 (33) with the following settings: LEADING:3, TRAILING:3, SLIDINGWINDOW:4:20, and MINLEN:36. Processed reads were then imported into QIIME 2 (Quantitative Insights Into Microbial Ecology 2)(34), and all further processing was performed using the MOSHPIT toolkit for shotgun metagenome analysis (35). Bowtie2 (36) and SAMtools (37) were used to align the reads against the human genome reference consortium (GRCh38) (38) to remove host sequences and retain nonhost (unmapped) reads for subsequent analysis, which resulted in a sum of 27,703,760 filtered reads across the data set.

### Microbiome taxonomy, diversity, and antimicrobial resistance gene profiling

The high-quality and filtered reads were taxonomically classified using Kraken2 (39) using the Standard database. This was followed by abundance re-estimation using Bracken (40) to enhance taxonomic profiling accuracy. Rarefied reads subsampled to the minimum sample read depth, in this case, 346,297 taxonomically annotated sequences per sample (excluding the unclassified portion), were compared on species-level dissimilarity based on beta diversity metrics, including Aitchison, Bray–Curtis, and Jaccard distances computed in QIIME 2. Additionally, alpha diversity metrics such as Shannon diversity and observed features (community richness) were also computed in QIIME 2.

To assess antimicrobial resistance (AMR) potential, reads were annotated using the Comprehensive Antibiotic Resistance Database (CARD) (41) in QIIME 2 using the q2-amr plugin (https://github.com/bokulich-lab/q2-amr). This involved using the Resistance Gene Identifier (RGI) application to predict antibiotic resistomes from protein or nucleotide data based on homology and single-nucleotide polymorphism (SNP) models. For validation using a database tailored to clinical applications, we used q2-amrfinderplus (https://github.com/bokulich-lab/q2-amrfinderplus) to annotate the assembled contigs with AMRFinderPlus (42).

### Metagenomic assembly and contig quality assessment

Trimmed and filtered reads from the individual samples were assembled into contigs using MEGAHIT (43) with default parameters using the QIIME 2 q2-assembly plugin (https://github.com/bokulich-lab/q2-assembly). Contig quality was assessed using metaQUAST (44) to calculate basic statistics such as assembly length, N50 values, and L50 metrics.

### Statistical analysis

Principal coordinate analysis (PCoA) was conducted using Aitchison, Bray–Curtis, and Jaccard distances in QIIME 2. To compare the mean Aitchison dissimilarity between different time points, the Wilcoxon rank-sum test was applied using the vegan package (v2.6–4) in RStudio (v2024.04.0 + 735), with $P$-values adjusted for multiple comparisons using the Bonferroni method. Additionally, PERMANOVA was performed with the vegan package (v2.6–4), utilizing the default number of permutations ($n = 999$). Pearson correlations were employed to assess the relationship between numerical or transformed metadata variables and principal components (PC1 and PC2) derived from the distance matrices. The Wilcoxon rank-sum test with Bonferroni correction was also used to evaluate differences in alpha diversity metrics, specifically Shannon diversity and observed features, across different time points. Differentially abundant analysis was performed at both species and genus levels using ANCOM-BC (45), with a minimum relative abundance of 1% across all samples.

To assess the influence of temporal versus inter-individual variation, we used the q2-sample-classifier plugin (46) (https://github.com/qiime2/q2-sample-classifier) in

QIIME 2 to train a random forest classifier to predict sample time points based on stool microbiome profiles. The analysis employed standard parameters, including an 80/20 split for training and testing and 100 trees in the random forest model, with performance evaluated through stratified k-fold cross-validation to determine the model's accuracy in predicting sample time points.

Linear mixed-effects (LME) models were employed to assess temporal trends in the microbial community composition and diversity while accounting for repeated measures within the same subjects over different time points and were performed using the QIIME 2 q2-longitudinal plugin (47) (https://github.com/qiime2/q2-longitudinal).

For constructing the heatmap and line-plots for antimicrobial resistance ontology (ARO) terms, normalized reads representing the number of sequence reads that were fully mapped to the reference sequence without any gaps were used to ensure accurate ARO term assignment. Mapped reads per sample were normalized for sequencing depth differences and to reduce sequencing bias by dividing the completely mapped reads by each sample's total read count and multiplying by 1,000,000 to obtain reads per million (RPM). These RPM values were then log-transformed to account for the variation in read ranges. The top 50 most abundant ARO terms were plotted in a heatmap using ggplot2 (v3.5.1) in RStudio (v2024.04.0 + 735). Similarly, presence/absence heatmaps were generated for AMR-conferring genes identified with RGI and AMRFinderPlus. For RGI, the presence/absence was determined based on reads mapping 100% to the reference database, while for AMRFinderPlus, it was based on contigs with 100% mapping. To maintain consistency, the analysis was restricted to the top 50 most frequent AMR genes identified by each tool.

Statistical analysis based on the Wilcoxon rank-sum test with Bonferroni correction of the N50 and L50 metrics was conducted to assess the quality of the metagenomic assemblies. These metrics were calculated and compared across the different storage time points to evaluate any changes in assembly quality.

## ACKNOWLEDGMENTS

We would like to acknowledge Silvio Ghezzi for fecal sample collection and transport and William Jakob for shotgun metagenomic sequencing data generation. We thank Dr. Michal Ziemski for his support on shotgun metagenomic sequencing analysis and Dr. Anton Lavrienko for his valuable comments on the manuscript.

This research was funded by a grant from the Swiss National Science Foundation (PZPGP3_193140).

## AUTHOR AFFILIATIONS

[1]Department of Community Health, Faculty of Science and Medicine, University of Fribourg, Fribourg, Switzerland
[2]Laboratory of Food Systems Biotechnology, Department of Health Sciences and Technology, ETH Zurich, Zurich, Switzerland
[3]Department of Paediatrics, Fribourg Hospital, Fribourg, Switzerland
[4]Infectious Diseases Research Group, Murdoch Children's Research Institute, Parkville, Australia
[5]Department of Paediatrics, The University of Melbourne, Parkville, Australia

## AUTHOR ORCIDs

Paula Momo Cabrera  http://orcid.org/0000-0002-6450-3075
Nicholas A. Bokulich  http://orcid.org/0000-0002-1784-8935
Petra Zimmermann  http://orcid.org/0000-0002-2388-4318

## FUNDING

| Funder | Grant(s) | Author(s) |
|---|---|---|
| Swiss National Science Foundation | PZPGP3_193140 | Petra Zimmermann |

## AUTHOR CONTRIBUTIONS

Paula Momo Cabrera, Data curation, Formal analysis, Investigation, Resources, Software, Visualization, Writing – original draft, Writing – review and editing | Nicholas A. Bokulich, Resources, Software, Supervision, Writing – review and editing | Petra Zimmermann, Conceptualization, Funding acquisition, Methodology, Project administration, Resources, Supervision, Writing – review and editing

## DATA AVAILABILITY

The sequencing raw data are deposited on the European Nucleotide Archive (ENA) with accession no. PRJEB79382. A STORMS (Strengthening The Organizing and Reporting of Microbiome Studies) checklist is available at DOI:10.5281/zenodo.13460391.

## ETHICAL APPROVAL

This study has been approved by the Commission cantonale d'éthique de la recherche sur l'être humain (The Cantonal Commission for Ethics in Research on Human Beings (CER-VD)) of the canton of Vaud, Switzerland (#2019–01567).

## ADDITIONAL FILES

The following material is available online.

### Supplemental Material

**Supplemental material (Spectrum02278-24-s0001.docx).** Figures S1 to S4; Tables S1 to S3.

### Open Peer Review

**PEER REVIEW HISTORY (review-history.pdf).** An accounting of the reviewer comments and feedback.

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
