## [Reviewer comments · Microbiology Spectrum]

Microbiology Spectrum

Evaluating stool microbiome integrity after domestic freezer storage using whole metagenome sequencing, genome assembly, and antimicrobial resistance gene analysis

Paula Momo Cabrera, Nicholas Bokulich, and Petra Zimmermann

Corresponding Author(s): Petra Zimmermann, Universite de Fribourg

Review Timeline:

Submission Date:	September 12, 2024
Editorial Decision:	October 25, 2024
Revision Received:	December 16, 2024
Editorial Decision:	December 29, 2024
Revision Received:	December 30, 2024
Accepted:	January 4, 2025

Editor: Jasna Kovac

Reviewer(s): Disclosure of reviewer identity is with reference to reviewer comments included in decision letter(s). The following individuals involved in review of your submission have agreed to reveal their identity: Elliot S Friedman (Reviewer #1); Maria Ingrid Cecilia Rubin (Reviewer #2)

Transaction Report:

DOI: <https://doi.org/10.1128/spectrum.02278-24>

Re: Spectrum02278-24 (Evaluating stool microbiome integrity after domestic freezer storage using whole metagenome sequencing, genome assembly, and antimicrobial resistance gene detection)

Dear Prof. Petra Zimmermann:

Thank you for the privilege of reviewing your work. Below you will find reviewers' comments and instructions for revision from the Spectrum editorial office.

Please thoroughly address reviewers' comments, make the sequencing data publicly available, and return the revised manuscript within 60 days; if you cannot complete the modification within this time period, please contact me. If you do not wish to modify the manuscript and prefer to submit it to another journal, notify me immediately so that the manuscript may be formally withdrawn from consideration by Spectrum.

Revision Guidelines

Sincerely,
Jasna Kovac
Editor
Microbiology Spectrum

Reviewer #1 (Comments for the Author):

1. Are the studies cited in lines 72-80 based on amplicon sequencing or shotgun metagenomic sequencing? Given the statement made in line 47, this is important to know.
2. Figure 1A. The figure legend infers that the time point ((0W = week 0, 1W = week 1, 2M = month 2, 6M = month 6) is shown in the figure but it is not. Should it be? Also, please find a better combination of colors/shapes to distinguish subjects. It is not possible to distinguish using current visuals (e.g., S17 and S03).

3. Figure 1B. The axes are missing. What is the thicker black line? What statistics are used? It is unclear based on the figure legend whether the statement about Wilcoxon rank-sum tests applies to Figure 1D only or Figure 1B-1D.
4. Figure 1C. Axes? Thick black line? Stats?
5. Figure 1D. What is mean distance? Is it the mean of all samples to all other samples, so $n = 20!$? If so, what do the individual dots indicate? Please clarify.
6. Figure 2A. Top 20 species based on mean relative abundance?
7. Figure 2B. Missing axes.
8. Figure 3A. The figure legend is exactly the same as Figure 1A, but the figure is different. Please explain.
9. Figure 3C. Please consider using log scale for y-axes.
10. Figure S1. Same comments about colors and labels as Figure 1A.
11. Figure S3. What is the difference between age (months) and age (years)? This is unclear.
12. Please provide brief overview of the study/samples at the beginning of the results, since the methods come after in this journal. Reader needs to know time points, study design, etc.
13. Line 127. Please clarify that you mean age of the subject?
14. Line 147. $>1%$ in any sample or $>1%$ average?
15. Line 151-152 "are likely attributable to inherent sample heterogeneity rather than storage conditions". Do you have any evidence or have you performed any analysis of this? I do not see it. Thus this statement should be removed.
16. Line 171 should be 'in contrast' not 'i contrast'.
17. I am unclear why the focus on AMR gene profiles over time rather than a global view of the metagenome, since the study is focused on sample storage methods. Why not perform a similar analysis as was done with taxonomy on the metagenome to determine if storage has an effect? It is unclear to me the relationship between this AMR gene analysis and the rest of the study.
18. Line 203 'shotgun' not 'gun'.
19. No records found for PRJEB79382 at ENA.

Reviewer #2 (Comments for the Author):

Please see Reviewer comments attached below

Reviewers comments

Thank you for a very interesting article on a highly important topic. For longitudinal analyses of fecal microbiome, it would be very beneficial to have the option of home-storage.

Major comments

In the discussion, lines 218-224, you mention that there were temporal fluctuations in specific taxa. For a measure of temporal variation, you have used various diversity indexes and AMR genes, but maybe the relative abundance that is quite fluctuating is actually a sign of that there are more temporal changes present than can be explained by looking only at the other metrics (Alpha-and Beta-diversity and AMR-genes). As this is difficult to measure statistically, I think it is worth noting that this might be more of a limitation to your study. Also, could one not measure the absolute abundance for a more statistically valuable quantity?

Furthermore, the sparsity of AMR-genes in your study, is probably due to the fact that these are all healthy children under the age of 4 from Switzerland. One would therefore not expect a great number of AMR-genes. It would therefore be very interesting to look at adults and especially elderly people, who have attained more AMR-genes, to more accurately evaluate the dynamics of AMR-genes. It should be added to the discussion, that a longitudinal monitoring of AMR-genes, might require other cohorts.

Lines 86-87: when you discuss the other studies here, it could be good to mention if they were well-powered as they showed different results.

Lines 108-109: it is hardly surprising that interindividual variation is greater than variations due storage duration. This has been described in abundance, and is not really a comparable parameter in this case.

Lines 167-170: it would be good to run also the AMR+Finder (AMRFinderPlus - Pathogen Detection - NCBI) as the one used in your paper, CARD, has some limitations. Especially as you hardly have any resistance genes it would be fine to compare two different resistance finder methods.

Lines 253-255: maybe also add, that apart from the study being very small, it should possibly contain other participant/patient cohorts, as mentioned above. Also looking at patients with specific gastrointestinal diseases, C.diff, patients colonized with multi-resistant organisms etc. Especially the latter could be of interest regarding fluctuation in AMR-genes.

Line 273-275: were these home-freezers all frost-free, that is, did they all have cycles of automatic frost removal as described on lines 88-89. Or were some of them constantly at -18°C? This could be added to the Methods section.

Minor comments

Line 39" under 4 years", could be good to add "of age"

Line 203: add "shot" before gun

Point-by-point response to editor and referees' comments

We thank the editor and reviewers for their appreciation and the constructive comments. We corrected the manuscript following all suggestions.

Reviewer #1 (Comments for the Author):

1. Are the studies cited in lines 72-80 based on amplicon sequencing or shotgun metagenomic sequencing? Given the statement made in line 47, this is important to know.

> We have addressed this point in the revised text. To clarify, all the studies mentioned in lines 72-80 were done using 16S rRNA amplicon sequencing. This distinction is critical given the context provided in line 47, and we have ensured this is explicitly stated in the manuscript to avoid ambiguity. Thank you for bringing this to our attention.

2. Figure 1A. The figure legend infers that the time point ((0W = week 0, 1W = week 1, 2M = month 2, 6M = month 6) is shown in the figure but it is not. Should it be? Also, please find a better combination of colors/shapes to distinguish subjects. It is not possible to distinguish using current visuals (e.g., S17 and S03).

> We have revised Figure 1A to include clear labeling of the time points (0W = week 0, 1W = week 1, 2M = month 2, 6M = month 6) directly within the figure legend for better clarity. Additionally, we have updated the color combinations to ensure that all subjects, including S17 and S03, are easily distinguishable. These changes have been reflected in both the figure and the corresponding text for clarity.

3. Figure 1B. The axes are missing. What is the thicker black line? What statistics are used? It is unclear based on the figure legend whether the statement about Wilcoxon rank-sum tests applies to Figure 1D only or Figure 1B-1D.

> We have updated Figure 1B to include appropriately labeled axes, clarifying the missing details. Additionally, we have revised the figure legend to clearly explain that the thicker black line represents the mean value, with the shaded gray area indicating the standard deviation. Furthermore, we clarified that the Wilcoxon rank-sum test with Bonferroni correction was applied to assess differences in panels B, C, and D, and no significant differences (ns) were observed across time points.

4. Figure 1C. Axes? Thick black line? Stats?

> We have updated Figure 1C as explained above for Figure 1B.

5. Figure 1D. What is mean distance? Is it the mean of all samples to all other samples, so $n = 20!$? If so, what do the individual dots indicate? Please clarify.

> We have clarified this in the figure legend. The mean Aitchison distance shown in Figure 1D represents the average pairwise distance for all samples within each time point ($n = 20$). Individual dots correspond to the pairwise distances calculated for each sample at that time point.

6. Figure 2A. Top 20 species based on mean relative abundance?

> *We have clarified this in the figure legend. We indeed refer to top 20 most abundant species across different time points.*

7. Figure 2B. Missing axes.

> *We have added visible axes to Figure 2B.*

8. Figure 3A. The figure legend is exactly the same as Figure 1A, but the figure is different. Please explain.

> *We have corrected Figure 3A legend. It shows beta diversity based on the Jaccard distance matrix, calculated from antimicrobial resistance (AMR)-conferring genes.*

9. Figure 3C. Please consider using log scale for y-axes.

> *We thank the reviewer for the suggestion to use a log scale for the y-axes in Figure 3C. To address this, we have applied a pseudo-log transformation (\log_{1p} , $\log(1 + x)$) to the y-axis. This approach provides the benefits of a logarithmic transformation, particularly for better visualizing trends in data with a wide range of values, while avoiding issues with zeros or very small values that are present in the dataset. The pseudo-log transformation ensures that all data points are retained and that trends for both low and high relative abundances can be appropriately represented. We have also updated the figure legend to reflect this change.*

10. Figure S1. Same comments about colors and labels as Figure 1A.

> *We have updated Figure S1 as explained above for Figure 1A.*

11. Figure S3. What is the different between age (months) and age (years)? This is unclear.

> *Age in months (age (months)) provides a more precise measure of the children's age, while age in years (age (years)) offers a broader, rounded approximation. We have clarified this distinction in the text.*

12. Please provide brief overview of the study/samples at the beginning of the results, since the methods come after in this journal. Reader needs to know time points, study design, etc.

> *We have included a brief overview of the study design, sample collection, and/or time points at the beginning of the results section to provide readers with the necessary context before delving into the findings.*

13. Line 127. Please clarify that you mean age of the subject?

> *We have now clarified this in the text that we refer to children's age.*

14. Line 147. >1% in any sample or >1% average?

> *We have adapted this in the text as:* "A total of 115 species were detected at or above 1% relative abundance amongst all samples"

15. Line 151-152 "are likely attributable to inherent sample heterogeneity rather than storage conditions". Do you have any evidence or have you performed any analysis of this? I do not see it. Thus this statement should be removed.

> *We have removed this from the text.*

16. Line 171 should be 'in contrast' not 'i contrast'.

>*We have corrected this in the text.*

17. I am unclear why the focus on AMR gene profiles over time rather than a global view of the metagenome, since the study is focused on sample storage methods. Why not perform a similar analysis as was done with taxonomy on the metagenome to determine if storage has an effect? It is unclear to me the relationship between this AMR gene analysis and the rest of the study.

> *AMR gene profiling in this study serves as a practical case study to evaluate the stability and DNA integrity of stool samples over time, complementing analyses like taxonomy or metagenome assembly quality. Focusing on AMR genes is particularly relevant because their profiling is a critical and growing application in shotgun metagenomic data. By incorporating this analysis, the study aligns with "real-world" research priorities while ensuring the methods tested are applicable to studies investigating AMR, which is an increasingly significant area of microbiome research.*

18. Line 203 'shotgun' not 'gun'.

>*We have corrected this in the text.*

19. No records found for PRJEB79382 at ENA.

> *We have submitted the raw shotgun data together with the metadata to the public repository European Nucleotide Archive (ENA) with accession number PRJEB79382 and made it publicly available.*

Reviewer #2 (Comments for the Author):

1. Thank you for a very interesting article on a highly important topic. For longitudinal analyses of fecal microbiome, it would be very beneficial to have the option of home-storage.

> *We thank the reviewer for their appreciation and the constructive comments.*

2. In the discussion, lines 218-224, you mention that there were temporal fluctuations in specific taxa. For a measure of temporal variation, you have used various diversity indexes and AMR genes, but maybe the relative abundance that is quite fluctuating is actually a sign of that there are more temporal changes present than can be

explained by looking only at the other metrics (Alpha- and Beta-diversity and AMR-genes). As this is difficult to measure statistically, I think it is worth noting that this might be more of a limitation to your study. Also, could one not measure the absolute abundance for a more statistically valuable quantity?

>We agree that fluctuations in relative abundance could indeed suggest temporal variation that might not be fully captured by diversity indices (Alpha- and Beta-diversity) or AMR gene dynamics alone. However, in our study, we performed Analysis of Composition of Microbiomes (ANCOM-BC) to identify differentially abundant taxa at both the species and genus levels. This method accounts for compositional effects inherent to relative abundance data and is statistically robust for detecting changes across conditions or timepoints. Despite employing this rigorous approach, we did not find any taxa that were significantly different over time, either at the species or genus level. This suggests that, while there may be fluctuations in relative abundance, these fluctuations are not statistically significant when corrected for multiple comparisons.

Regarding the suggestion to measure absolute abundance: while this could provide additional insights, obtaining absolute abundance data requires either cell quantification (e.g., qPCR, flow cytometry) or spike-in standards during sequencing, which were not part of our experimental design.

We acknowledge that temporal fluctuations in relative abundance might reflect ecological dynamics that are not fully captured by the metrics we used. As such, we have added a statement to the discussion noting this as a potential limitation of our study. Future work incorporating absolute abundance measurements and additional time-series analysis methods could provide deeper insights into these temporal dynamics.

3. Furthermore, the sparsity of AMR-genes in your study, is probably due to the fact that these are all healthy children under the age of 4 from Switzerland. One would therefore not expect a great number of AMR-genes. It would therefore be very interesting to look at adults and especially elderly people, who have attained more AMR-genes, to more accurately evaluate the dynamics of AMR-genes. It should be added to the discussion, that a longitudinal monitoring of AMR-genes, might require other cohorts

>We thank the reviewer for the comment. Several studies indicate that the gut can have a higher abundance of ARGs in early life compared to adulthood(1,2). This is attributed to early colonization by antibiotic-resistant bacteria acquired from maternal and environmental sources, with ARG levels gradually declining as the microbiome matures. We have incorporated this evidence into the discussion. Additionally, we have emphasized the need for longitudinal studies across diverse cohorts, including adults and elderly populations, to comprehensively assess the dynamics of AMR genes over time and across life stages.

4. Lines 86-87: when you discuss the other studies here, it could be good to mention if they were well-powered as they showed different results.

> We have included the sample sizes of the referenced studies (n = 11, 4, and 3) and noted that differences in findings may reflect variability in sample sizes and study designs.

5. Lines 108-109: it is hardly surprising that interindividual variation is greater than variations due storage duration. This has been described in abundance and is not really a comparable parameter in this case.

> We have revised the paragraph to focus specifically on the impact of storage duration, omitting the mention of inter-individual variation as a comparison point. The updated text now highlights that storage at home freezers had minimal impact on the microbial community structure, as no clustering by storage duration was observed.

6. Lines 167-170: it would be good to run also the AMR+Finder (AMRFinderPlus - Pathogen Detection - NCBI) as the one used in your paper, CARD, has some limitations. Especially as you hardly have any resistance genes it would be fine to compare two different resistance finder methods.

> We thank the reviewer for the valuable suggestion to compare the Comprehensive Antibiotic Resistance Database (CARD) with AMRFinderPlus. We conducted an analysis using both tools and included Figure S3, which visualizes the presence/absence of the top 50 most frequent AMR-conferring genes detected by each tool. Our findings demonstrate that both tools identified clinically relevant AMR genes, such as *tet(Q)*, *erm(B)*, and *cfxA*, and showed consistent detection of these genes across time points (0W, 1W, 2M, and 6M), confirming the longitudinal stability of these AMR profiles. However, as anticipated, differences between the tools were observed. AMRFinderPlus prioritized clinically significant resistance genes, while CARD (via RGI) annotated a broader range of resistance mechanisms, including efflux pump-associated genes like *mdtA* and *acrB*. There is also a known difference in the number of genes present in each database, which contributes to the observed variations in annotations (see Venn diagram below).

7. Lines 253-255: maybe also add, that apart from the study being very small, it should possibly contain other participant/patient cohorts, as mentioned above. Also looking at patients with specific gastrointestinal diseases, *C.diff*, patients colonized with multi-resistant organisms etc. Especially the latter could be of interest regarding fluctuation in AMR-genes.

> We have revised the limitations section to address the need for including diverse participant cohorts in future research. Specifically, we now highlight the importance of expanding microbiome studies to include individuals with gastrointestinal diseases (e.g., *Clostridioides difficile* infections) and those colonized by multi-resistant organisms. These populations are particularly relevant due to their higher abundances and diversity of AMR genes, which could provide valuable insights into the stability of microbial composition and AMR gene profiles under different storage conditions.

8. Line 273-275: were these home-freezers all frost-free, that is, did they all have cycles of automatic frost removal as described on lines 88-89. Or were some of them constantly at -18°C? This could be added to the Methods section.

> *We have added freezer models used in the study to the Methods section. Specifically, we now state that the domestic freezers included several models (e.g., V-ZUG Classic eco, Bosch Serie 6 NoFrost, Indesit Class TZAAA10.1, Liebherr MedLine, and Electrolux SG 164). The Bosch Serie 6 NoFrost and Liebherr MedLine freezers were frost-free models with automatic defrost cycles, while the V-ZUG, Indesit, and Electrolux models required manual defrosting.*

9. Line 39" under 4 years", could be good to add "of age"

> *We included the suggested change.*

10. Line 203: add "shot" before gun

> *We included the suggested change.*

References

1. Yassour M, Jason E, Hogstrom LJ, Arthur TD, Tripathi S, Siljander H, et al. Strain-Level Analysis of Mother-to-Child Bacterial Transmission during the First Few Months of Life. *Cell Host & Microbe*. 2018 Jul 11;24(1):146-154.e4.
2. Moore AM, Ahmadi S, Patel S, Gibson MK, Wang B, Ndao IM, et al. Gut resistome development in healthy twin pairs in the first year of life. *Microbiome*. 2015 Jun 25;3(1):27.

Re: Spectrum02278-24R1 (Evaluating stool microbiome integrity after domestic freezer storage using whole metagenome sequencing, genome assembly, and antimicrobial resistance gene analysis)

Dear Prof. Petra Zimmermann:

Thank you for the privilege of reviewing your work. Based on your responses to the reviewers' comments, I recommend a minor revision:

- Please revise the Figure 1 legend to match the description of individual figure panels. Specifically, the figure legend describes the Shannon diversity index over time in panel B, but this graph is labeled as panel C in the figure. Furthermore, there is no panel D label in Figure 1.
- In Figure S1, it is not clear which samples were collected in which week, although the response to reviewers' comments indicates that this has been addressed and revised.
- Please ensure that comment #13 of the Reviewer 1 is addressed in the revised manuscript.

Revision Guidelines

Sincerely,
Jasna Kovac
Editor
Microbiology Spectrum

Point-by-point response to editor and referees' comments

We thank the editor and reviewers for their appreciation and the constructive comments. We corrected the manuscript following all suggestions.

1. Please revise the Figure 1 legend to match the description of individual figure panels. Specifically, the figure legend describes the Shannon diversity index over time in panel B, but this graph is labeled as panel C in the figure. Furthermore, there is no panel D label in Figure 1.

> We have corrected the legend of Figure 1.

2. In Figure S1, it is not clear which samples were collected in which week, although the response to reviewers' comments indicates that this has been addressed and revised.

> We have revised the legend of Figure S1. As the panel is focused on illustrating beta diversity across subjects rather than time points, we have confirmed that time point distinction is not necessary for this panel and have updated the figure legend accordingly. These changes have been reflected in both the figure and the corresponding text for clarity.

3. Please ensure that comment #13 of the Reviewer 1 is addressed in the revised manuscript.

> We have clarified this in the text that we refer to children's age (see line 133 of the cleaned manuscript).

Re: Spectrum02278-24R2 (Evaluating stool microbiome integrity after domestic freezer storage using whole metagenome sequencing, genome assembly, and antimicrobial resistance gene analysis)

Dear Dr. Zimmermann:

Your manuscript has been accepted, and I am forwarding it to the ASM production staff for publication. Your paper will first be checked to make sure all elements meet the technical requirements. ASM staff will contact you if anything needs to be revised before copyediting and production can begin. Otherwise, you will be notified when your proofs are ready to be viewed.

Sincerely,
Jasna Kovac
Editor
Microbiology Spectrum